# Higher harmonics in planar Hall effect induced by cluster magnetic multipoles

Jeongkeun Song[1,2], Taekoo Oh [1,2,3], Eun Kyo Ko[1,2], Ji Hye Lee[1,2], Woo Jin Kim [4,5], Yangyu Zhu[6], Bohm-Jung Yang [1,2,3], Yangyang Li [6] ✉ & Tae Won Noh [1,2] ✉

Antiferromagnetic (AFM) materials are attracting tremendous attention due to their spintronic applications and associated novel topological phenomena. However, detecting and identifying the spin configurations in AFM materials are quite challenging due to the absence of net magnetization. Herein, we report the practicality of utilizing the planar Hall effect (PHE) to detect and distinguish "cluster magnetic multipoles" in AFM $Nd_2Ir_2O_7$ (NIO-227) fully strained films. By imposing compressive strain on the spin structure of NIO-227, we artificially induced cluster magnetic multipoles, namely dipoles and $A_2$- and $T_1$-octupoles. Importantly, under magnetic field rotation, each magnetic multipole exhibits distinctive harmonics of the PHE oscillation. Moreover, the planar Hall conductivity has a nonlinear magnetic field dependence, which can be attributed to the magnetic response of the cluster magnetic octupoles. Our work provides a strategy for identifying cluster magnetic multipoles in AFM systems and would promote octupole-based AFM spintronics.

Antiferromagnetic (AFM) materials have attracted considerable interest as promising materials for next-generation spintronics[1–5] and novel topological phenomena[6–11]. This interest is largely due to the fact that the AFM spin arrangement produces zero net magnetization, leading to an absence of stray fields and insensitivity to external magnetic fields[12,13]. Additionally, the spin precession for AFM order is much faster than that in ferromagnets[14]. The resonance frequencies in antiferromagnets are in the terahertz range[15–17], whereas those of ferromagnets are in the gigahertz range. In parallel with AFM spintronics, antiferromagnets with topologically protected states have become another focus of recent interest. As AFM spin order breaks the time-reversal symmetry (TRS) or inversion symmetry, various topologically nontrivial states including the Weyl semimetal[6–8], axion insulator[9,10], and Möbius insulator[11] can theoretically emerge.

To develop novel AFM spintronics and investigate topological states in AFM materials, understanding the relationship between the spin texture and emergent phenomena in AFM materials is essential. This relationship can be theoretically described by the recently introduced cluster multipole theory (CMT)[18]. In CMT[18], magnetic structures are classified as "cluster multipoles" (dipole and additional high-rank multipoles) according to irreducible representations of the crystallographic point group. These cluster multipoles can be regarded as order parameters that reflect symmetry breaking in AFM materials. In particular, some cluster magnetic octupole (CMO) (i.e., $T_1$ octupole) belongs to the same magnetic point group (−42'm') as the magnetic dipole[18], leading to symmetry breaking and generating nonvanishing Berry curvature. In fact, the analysis using CMOs has been recently extended to understanding the large response of AHE[18–21], magneto-optical signals[22], the SHE[23–25], the anomalous Nernst effect[26–28], and perpendicular magnetization[29,30] in AFM materials. However, due to the absence of net magnetization, experimental identification of the CMOs in AFM materials is very challenging. To date, it has been limited

[1]Center for Correlated Electron Systems, Institute for Basic Science (IBS), Seoul 08826, Korea. [2]Department of Physics and Astronomy, Seoul National University, Seoul 08826, Korea. [3]Center for Theoretical Physics (CTP), Seoul National University, Seoul 08826, Korea. [4]Stanford Institute for Materials and Energy Sciences, SLAC National Accelerator Laboratory, Menlo Park, CA 94025, USA. [5]Department of Applied Physics, Stanford University, Stanford, CA 94305, USA. [6]School of Physics, Shandong University, Jinan 250100, China. ✉e-mail: yangyang.li@sdu.edu.cn; twnoh@snu.ac.kr

to X-ray magnetic circular dichroism measurements[31] and the neutron scattering technique[32].

The planar Hall effect (PHE) has been considered a method of probing the physical properties of materials such as magnetism and topology. The PHE corresponds to the development of a Hall voltage when electric and magnetic fields are coplanar, which is different from the usual Hall effect where they are perpendicular to each other. Initially, the PHE was observed in ferromagnetic systems[33–36], detecting the anisotropic magnetization of ferromagnetic materials. Additionally, the PHE has recently been in the spotlight due to its role in detecting topological characteristics such as the chirality arising from Weyl fermions in magnetic Weyl semimetals[37–40]. The associated PHE in both ferromagnets and Weyl semimetals exhibit $\sin(2\phi)$ or second harmonic PHE oscillations. In contrast, higher harmonics PHE oscillations beyond second in topological systems[41–44] have been recently reported, in which higher harmonics PHE originated from additional unknown parameters in materials.

Among many kinds of AFM materials, the family of *5d* AFM materials $R_2Ir_2O_7$ (R: rare-earth ions: Eu, Y, Nd, and Pr) has received considerable attention due to a plethora of intriguing properties[6,7]. The crystal structure of NIO-227 is composed of Ir and Nd tetrahedral sublattices (Fig. 1a), which have a symmetry identical to that of the diamond lattice. Most $R_2Ir_2O_7$ bulk compounds exhibit an intriguing spin configuration called all-in-all-out (AIAO). According to CMT, the AIAO ordering of bulk $R_2Ir_2O_7$ is equivalent to the $A_2$-CMO[21,45]. This $A_2$-CMO breaks the TRS and generates nonvanishing Berry curvature, resulting in topological properties in momentum space such as the magnetic Weyl semimetal[6] and Axion insulator[6].

Here, we demonstrate the detection and identification of CMO in fully-strained AFM $Nd_2Ir_2O_7$ (NIO-227) film via the PHE. Under epitaxial strain, three kinds of cluster multipoles (dipoles and $A_2$- and $T_1$-CMOs) can be induced in the NIO-227 film. By rotating the magnetic field, distinctive harmonics in the PHE oscillation were observed. Specifically, the dipole induces the second harmonic due to longitudinal magnetization (Fig. 1c), whereas the $A_2$- and $T_1$-CMO induce fourth and sixth harmonics in the PHE oscillation, respectively. We demonstrate that the higher harmonics of the PHE oscillation for both CMOs originate from the magnetic response of the CMOs, called orthogonal magnetization (OM) (Fig.1d). Furthermore, cluster multipoles induce nonlinear magnetic field-dependent the planar Hall conductivity. This nonlinear magnetic field behavior of the planar Hall conductivity can be understood in terms of the magnetic properties of the cluster magnetic dipole and CMOs.

## Results

To investigate the magnetotransport properties of CMOs, we fabricated 15 nm fully strained NIO-227 films on a (111)-oriented yttria-stabilized zirconia (YSZ) substrate. More details on the growth are provided in the Methods section. Figure 2a shows the X-ray diffraction (XRD) pattern of a (111)-oriented NIO-227 thin film on a YSZ substrate. The strong odd-pair peaks in the XRD pattern exhibit excellent crystallization of the pyrochlore phase in the NIO film. Further analysis of the reciprocal space map (Fig. 2b) shows that the NIO-227 film is fully strained with a compressive strain of ~1%. As mentioned earlier, the presence of compressive strain in the NIO-227 film is crucial, as it modulates the AIAO ordering and develops $T_1$-CMOs. Figure 2c shows

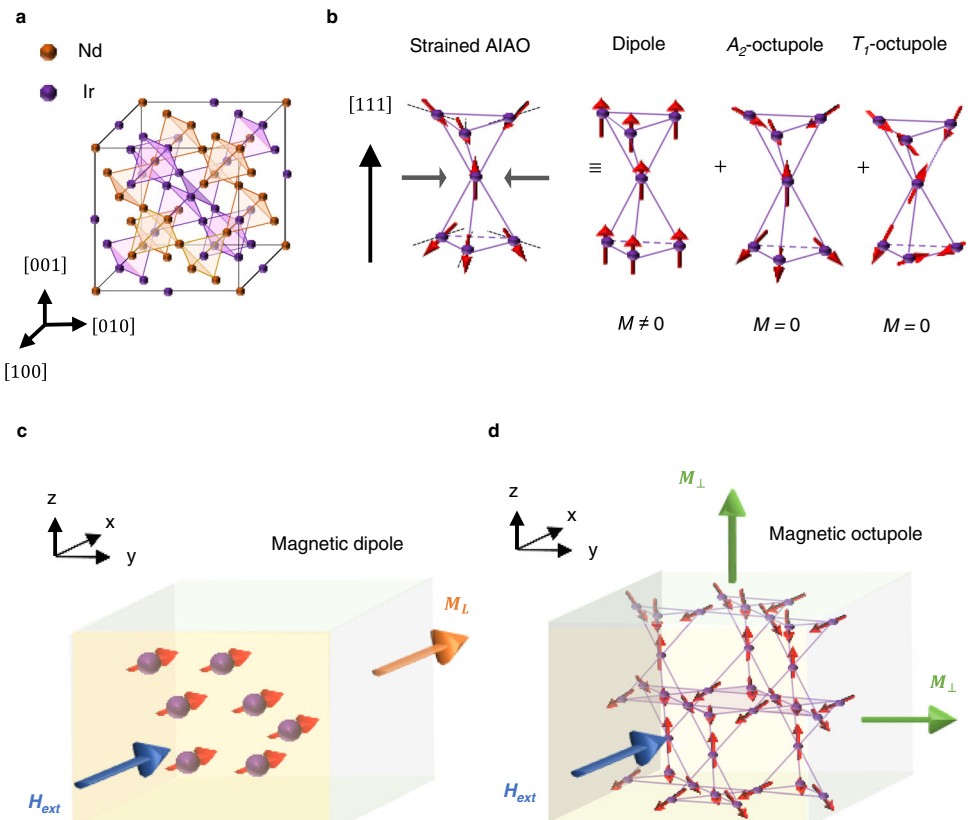

**Fig. 1 | Schematic diagram of strain-induced cluster magnetic multipoles and OM in Nd₂Ir₂O₇ thin films. a** Schematic view of the pyrochlore lattice structure of Nd₂Ir₂O₇. **b** When compressive strain is applied to Nd₂Ir₂O₇ along the [111] direction, the AIAO configuration of the magnetic spin experiences spin canting. Such canting can be represented by three distinctive magnetic multipoles (dipole, A₂-octupole, and T₁-octupole). Note that the dipole has finite magnetization, whereas

the A₂-octupole and T₁-octupole do not have magnetization. **c** Schematic diagram of the longitudinal magnetization induced by the dipole. When $H_{ext}$ is applied along the x-y plane, the spin direction is aligned in the direction of $H_{ext}$. **d** Schematic diagram of the OM induced by coupling between an external magnetic field ($H_{ext}$) and magnetic octupoles. The OM can be induced in two directions: normal to the surface and in-plane.

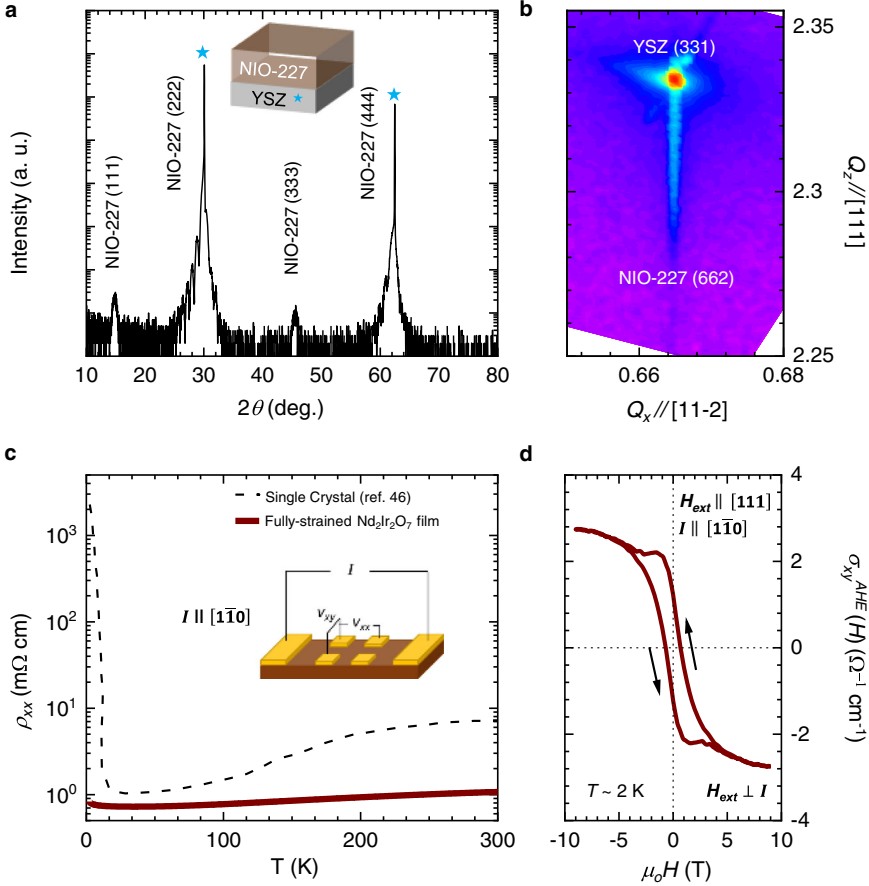

**Fig. 2 | Characterization of a Nd$_2$Ir$_2$O$_7$ thin film. a** High-resolution XRD pattern of a 15 nm thick Nd2Ir2O7 (NIO-227) thin film. The NIO-227 thin film was grown on a YSZ substrate. **b** Reciprocal space map of the NIO-227 film in the vicinity of the YSZ (331) reflection, indicating that the NIO-227 film is fully strained. **c** The plot of the longitudinal resistivity, $\rho_{xx}$, vs. temperature, $T$, of NIO-227 thin film and single crystal (ref. 46 in the main manuscript). **d** The AHE is measured by applying $H_{ext}$ along the [111] direction and $I$ along the [1$\bar{1}$0] direction at 2 K. A clear AHE without a magnetic field is the signature of the presence of the $T_1$-octupole in the NIO-227 thin film.

the longitudinal resistivity $\rho_{xx}$ $(T)$ of the NIO-227 thin film as a function of temperature without a magnetic field. Clearly, with decreasing $T$, the NIO-227 film exhibits stronger semimetallic behavior than that of bulk NIO-227. At $T \sim 2$ K, our NIO-227 film has a $\rho_{xx}$ of ~2.28 mΩ cm, whereas that of bulk NIO-227 is ~ 2.20 × 10$^3$ mΩ cm[46]. With compressive strain, the valence and conduction bands move in the NIO-227 film. The valence and conduction bands cross near the Fermi level, resulting in the development of electron and hole pockets[21]. These previous model calculations explain the enhancement of the conductivity in the NIO-227 thin film compared to the bulk.

Moreover, the AHE ($\sigma_{xy}^{AHE}$ $(H)$) of the NIO-227 film was measured with $H_{ext}$ applied along the [111] direction below 30 K. As shown in Fig. 2d, $\sigma_{xy}^{AHE}$ $(H)$ at 2 K shows a hysteric feature with a finite value at 0 T, known as the spontaneous Hall effect[21]. In our strained NIO-227 thin film, the AHE and spontaneous Hall effect developed below 30 and 15 K, respectively. The value of the spontaneous Hall effect approached the maximum value at 2 K. More details are provided in Supplementary Materials (Supplementary Fig. 1 and Note 1). The appearance of AHE below 30 K was previously attributed to Ir spin ordering[21]. In contrast, the emergence of the spontaneous Hall effect without magnetization below 15 K in the NIO-227 film is known to be the nontrivial contribution of the $T_1$-CMO to the Berry curvature[21], which is amplified by Nd ordering through $f$-$d$ exchange interaction. The effect of Nd ordering can give an additional hysteric component on AHE[21] (Supplementary Fig. 2). In our 15 nm NIO-227 film, the spontaneous Hall effect is observed without magnetization, which is induced by $T_1$-CMO in the film (Supplementary Fig. 1c).

To detect CMOs in the NIO-227 film, we performed PHE measurements. As shown in Fig. 3a, the PHE ($\Delta\sigma_{xy}^{PHE}$ ($\phi$, $H = \pm 9$ T)) below 30 K was measured across the width of the sample by applying $I$ along the [1$\bar{1}$0] direction with the rotation of $H_{ext}$. The details about the experimental measurement and data process are provided in the "Methods" and Supplementary Materials (Supplementary Fig. 3 and Note 2), respectively. In the range of 15–30 K, the $\Delta\sigma_{xy}^{PHE}$ ($\phi$) curves in the NIO-227 film mostly exhibit sin (2$\phi$) oscillations (Fig. 3b). The sin (2$\phi$) behavior (second harmonic) of the $\Delta\sigma_{xy}^{PHE}$ ($\phi$) oscillation above 15 K can be understood as the magnetization from dipolar order. As shown in Fig. 1b, the NIO-227 film has a cluster dipole, which can be regarded as ferromagnetic ordering. This cluster dipole induces $M_L$ to be directed along $H_{ext}$, which can result in the second harmonic of the $\Delta\sigma_{xy}^{PHE}$ ($\phi$) oscillation. Thus, the second harmonic of the $\Delta\sigma_{xy}^{PHE}$ ($\phi$) oscillation above 15 K can be understood as the transverse voltage developed by the cluster dipole in the NIO-227 film under $H_{ext}$.

Below $T \sim 15$ K, the behavior of the $\Delta\sigma_{xy}^{PHE}$ ($\phi$) curves are complex but periodic, exhibiting oscillations of multiple harmonics. As shown in Fig. 3b, the $\Delta\sigma_{xy}^{PHE}$ ($\phi$) oscillation at 2 K cannot be explained only by the typical second harmonic induced by dipolar order. Such complex behavior of the $\Delta\sigma_{xy}^{PHE}$ ($\phi$) oscillation can be seen in the contour plot presented in Fig. 3c. Whereas the second harmonic of $\Delta\sigma_{xy}^{PHE}$ ($\phi$) (green region) exists in all temperature regions below 30 K, additional harmonics (red region) of the $\Delta\sigma_{xy}^{PHE}$ ($\phi$) oscillation appears below 15 K. As a result, the $\Delta\sigma_{xy}^{PHE}$ ($\phi$) curve exhibits oscillations of multiple harmonics below 15 K.

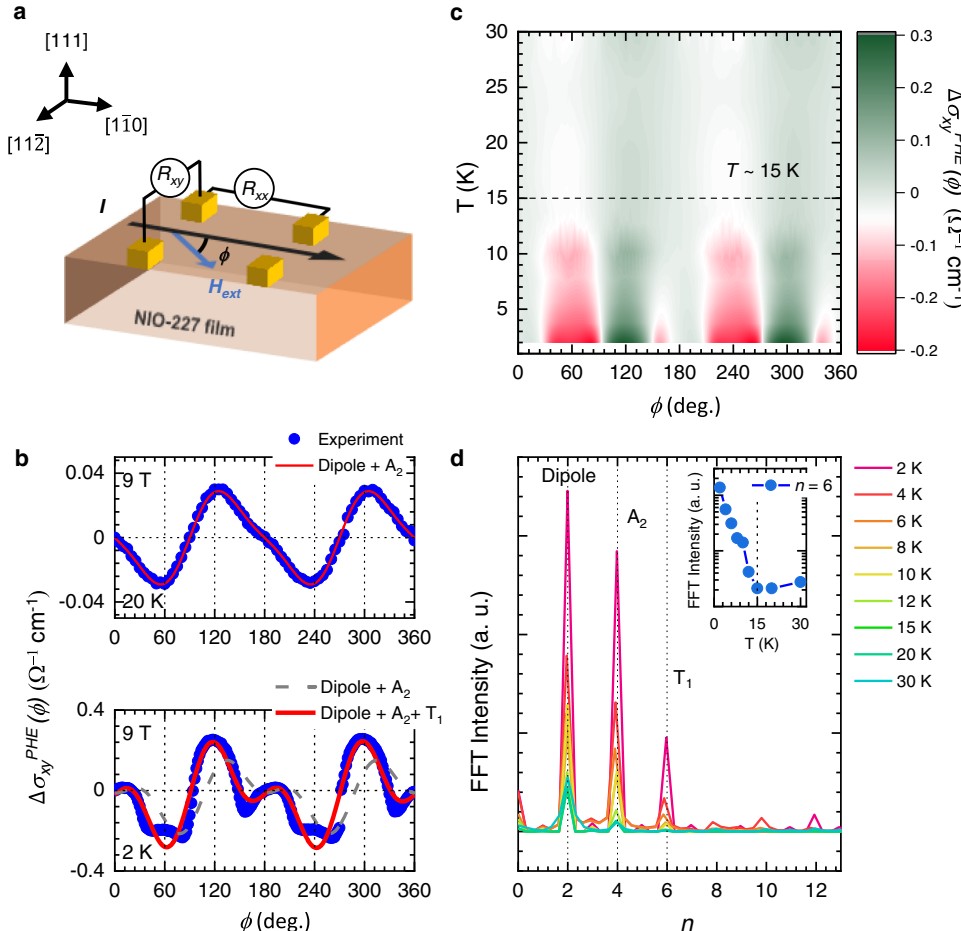

**Fig. 3 | Anomalous magnetic oscillation in the $\Delta\sigma_{xy}^{PHE}(\phi, H = \pm 9\,T)$ of the NIO-227 thin film. a** Schematic diagram of the $\Delta\sigma_{xy}^{PHE}(\phi)$ measurement geometry. In this geometry, $I$ is applied along the $[1\bar{1}0]$ direction and $H_{ext}$ is rotated within the sample plane. **b** $\Delta\sigma_{xy}^{PHE}(\phi)$ curves at 20 and 2 K. At 20 K, the $\Delta\sigma_{xy}^{PHE}(\phi)$ curve has a sin $(2\phi)$ oscillation. At 2 K, the $\Delta\sigma_{xy}^{PHE}(\phi)$ curve has an inhomogeneous oscillation, indicating a hidden magnetic origin. **c** Contour plot of all $\Delta\sigma_{xy}^{PHE}(\phi)$ curves below 30 K. A gradual appearance of the abovementioned peaks can be seen (red region). While

the peaks at $\phi = 120°$ and $300°$ (green region) exist above $T \sim 15$ K, the peaks near $\phi = 60°$, $180°$, $240°$, and $340°$ (red region) appear below $T \sim 15$ K. **d** FFT results of all $\Delta\sigma_{xy}^{PHE}(\phi)$ curves with different $T$. The FFT intensities of the second, fourth, and sixth harmonics are related to magnetization from the dipole, OM induced by the $A_2$-CMO, and OM induced by the $T_1$-CMO, respectively. The inset shows the temperature dependence of the sixth harmonic below 30 K. Clearly, the sixth harmonic order emerges at 15 K, which indicates the role of the $T_1$-CMO.

To obtain further insight into the anomalous behavior of $\Delta\sigma_{xy}^{PHE}(\phi)$ curves below 15 K, we performed a fast Fourier transform (FFT) of the measured $\Delta\sigma_{xy}^{PHE}(\phi)$ curves. The detailed FFT procedure is provided in Supplementary Materials (Supplementary Fig. 4). The harmonics $(n)$ dependency of the FFT intensity in the $\Delta\sigma_{xy}^{PHE}(\phi)$ oscillation is shown in Fig. 3d. At 30 K, the second harmonic always exists, which is induced by the cluster dipole. The additional higher harmonics appear below 30 K. The appearance of the fourth harmonic in the $\Delta\sigma_{xy}^{PHE}(\phi)$ oscillation has been attributed to the $A_2$-CMO-induced chiral anomaly in $Pr_2Ir_2O_7$ thin films[41]. Considering that both $Pr_2Ir_2O_7$ and NIO-227 films have the $A_2$-CMO, the observed fourth harmonic of the $\Delta\sigma_{xy}^{PHE}(\phi)$ oscillation in the NIO-227 film should originate from the $A_2$-CMO. However, the appearance of the sixth harmonic below 15 K (inset in Fig. 3d) cannot be explained by either the dipole or $A_2$-CMO.

To elucidate the appearance of the sixth harmonic in the $\Delta\sigma_{xy}^{PHE}(\phi)$ oscillation, we calculate $\Delta\sigma_{xy}^{PHE}$ from the dipole and $A_2$- and $T_1$-CMOs using the phenomenological model. Notably, the $A_2$-CMO induces an unusual magnetic response perpendicular to $H_{ext}$. This magnetic response was initially observed in a $Eu_2Ir_2O_7$ single crystal by torque magnetometry and defined as OM[30]. As mentioned above, the $T_1$-CMO is also expected to exhibit OM. Employing the expression[30] of the OM in $Eu_2Ir_2O_7$, we obtained the $\phi$ dependence of $\Delta\sigma_{xy}^{PHE}(\phi)$ for our system,

expressed as follows:

$$\Delta\sigma_{xy}^{PHE}(\phi) = \sigma_{xy}^{Dipole} + \sigma_{xy}^{A2} + \sigma_{xy}^{T1} = A\sin(2\phi) + B\sin(4\phi) + C\sin(6\phi) \quad (1)$$

where $A$ is the PHE coefficient of the dipole. $B$ and $C$ are the PHE coefficients of the OM induced by $A_2$- and $T_1$-CMOs, respectively. The detailed theoretical calculation is discussed in Supplementary Materials (Supplementary Figs. 5 and 6, and Note 3): Then, Eq. (1) can explain the appearance of peaks of the second, fourth, and sixth harmonics (Fig. 3d) by the $M_L$ induced by the dipole, OM induced by the $A_2$-CMO, and OM induced by the $T_1$-CMO, respectively. In the range of 15–30 K, the $\Delta\sigma_{xy}^{PHE}(\phi)$ oscillation can be fitted with the $M_L$ and OM induced by the dipole and $A_2$-CMO, respectively. The fitting of the $\Delta\sigma_{xy}^{PHE}(\phi)$ oscillation at 20 K without the sin $(6\phi)$ term in Eq. (1) is shown in Fig. 3b. This indicates that the cluster dipole and $A_2$-CMO of Ir spin ordering induce second and fourth harmonics of the $\Delta\sigma_{xy}^{PHE}(\phi)$ oscillation, respectively, while the contribution of the $T_1$-CMO is small in this temperature region.

In contrast, the $\Delta\sigma_{xy}^{PHE}(\phi)$ oscillation below 15 K can only be fitted by including the $T_1$-CMO contribution. As shown in Fig. 3b, Eq. (1), by including the sin $(6\phi)$ term from the $T_1$-CMO (red), well fits $\Delta\sigma_{xy}^{PHE}(\phi)$ at

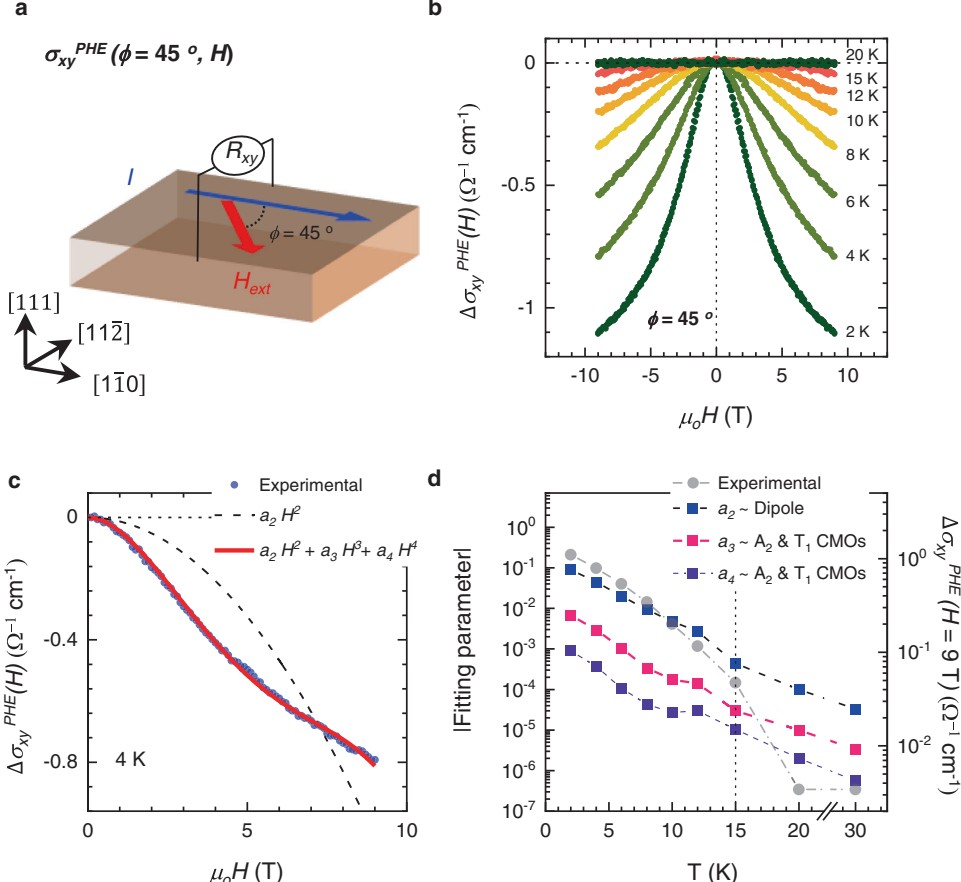

**Fig. 4 | Nonlinear magnetic behavior of the $\Delta\sigma_{xy}^{PHE}(\phi = 45°, H)$ of the NIO-227 thin film. a** Schematic diagram of the $\Delta\sigma_{xy}^{PHE}(H)$ measurement geometry. In this geometry, the angle between the current and magnetic field (±9 T) is 45°. Surprisingly, $\Delta\sigma_{xy}^{PHE}(H)$ shows $H_{ext}$ dependency below $T \sim 15$ K due to OM. **b** $\Delta\sigma_{xy}^{PHE}(H)$ curves as a function of $H_{ext}$ below $T \sim 20$ K. In the region of $15 K < T < 20$ K, the value of $\Delta\sigma_{xy}^{PHE}(H)$ is nearly zero. The slight deviation of $\Delta\sigma_{xy}^{PHE}(H)$ near $H_{ext} \sim ±9$ at $T \sim 15$ K is possibly due to dipolar order. **c** Logarithmic plot of the results of fitting $\Delta\sigma_{xy}^{PHE}(H)$ at $T \sim 4$ K

with $\Delta\sigma_{xy}^{PHE}(H) = a_2H^2 + a_3H^3 + a_4H^4$ (red line). The $a_2$, $a_3$, and $a_4$ terms correspond to longitudinal magnetization, OM from the $A_2$-CMO, and OM from the $T_1$-CMO, respectively. Note that without the $a_3$ and $a_4$ contributions (black dashed line), the experimental data (blue dots) do not fit well, indicating the role of OM induced by both the $A_2$- and $T_1$-octupoles. **d** The plot of extracted fitting parameters $a_2$, $a_3$, and $a_4$ and $\Delta\sigma_{xy}^{PHE}(H = 9 T)$ for the measured $T$.

2 K. The fitting results of $\Delta\sigma_{xy}^{PHE}(\phi)$ for measured temperature using Eq. (1) is provided in the Supplementary Materials (Supplementary Fig. 7). The temperature dependence of the fitting result is consistent with the appearance of the sixth harmonic peak below 15 K in the NIO-227 film. Furthermore, a similar temperature characteristic of $T_1$-CMOs is observed in not only AHE but also in anisotropic magnetoconductivity and out-of-plane rotation magnetoconductivity (Supplementary Figs. 8 and 9). The multiple harmonics of anisotropic magnetoconductivity and out-of-plane rotation magnetoconductivity are developed below 15 K. These complex behaviors can be explained by including the contribution of $T_1$-CMOs, which is amplified by Nd ordering through $f$-$d$ exchange interaction[21]. Therefore, we can confirm that the higher harmonics of $\Delta\sigma_{xy}^{PHE}(\phi)$ originate from $A_2$- and $T_1$-CMOs via OM.

Since the OM is the coupling effect between CMOs and $H_{ext}$ and is known to have an $H$-field dependence[30], the $\sigma_{xy}^{PHE}(H)$ of NIO-227 should exhibit behavior different from that of a ferromagnet. We measured $\Delta\sigma_{xy}^{PHE}(\phi = 45°, H)$ below 30 K by fixing $\phi = 45°$ between $I$ and $H_{ext}$ (Fig. 4a). $R_{xx}$ and $R_{xy}$ were simultaneously measured and symmetrized, and $\Delta\sigma_{xy}^{PHE}(H)$ was obtained (Supplementary Fig. 10). The $\Delta\sigma_{xy}^{PHE}(H)$ curves with respect to $H_{ext}$ below 20 K are shown in Fig. 4b. In the range of 15–20 K, the values of the $\Delta\sigma_{xy}^{PHE}(H)$ curves remain near zero at $H_{ext} = +9$ and $−9$T. However, below 15 K, the values of the $\Delta\sigma_{xy}^{PHE}(H)$ curves have finite values with increasing $H_{ext}$ and reach $\Delta\sigma_{xy}^{PHE}(H) = +9T \sim −1.103\ \Omega^{-1}\ cm^{-1}$ at 2 K.

By considering the OM induced by the $A_2$-CMO and $T_1$-CMO, we calculated $\Delta\sigma_{xy}^{PHE}(H)$ using the phenomenological model. Based on our calculations, the relation of $\Delta\sigma_{xy}^{PHE}(H)$ to $H_{ext}$ can be expressed as follows:

$$\Delta\sigma_{xy}^{PHE}(H) = a_2H^2 + a_3H^3 + a_4H^4 \qquad (2)$$

where $a_2$ is the coefficient of longitudinal magnetization induced by the dipole. Both $a_3$ and $a_4$ are the coefficients of OM induced by $A_2$- and $T_1$-CMOs. The details about the data process and theoretical calculation are provided in Supplementary Materials (Supplementary Note 4). Figure 4c shows the fitting results for $\Delta\sigma_{xy}^{PHE}(H)$ at a selected temperature of $\sim 4$ K with (red line) and without (black dashed line) the $H^3$ and $H^4$ terms. Note that the exact proportions of dipole and CMOs cannot be determined by experimental PHE due to the complexity behind transport phenomena[41], but we can get it theoretically (Supplementary Fig. 11). The fitting with the OM contribution explains the experimental results better than that with the dipole, demonstrating the development of OM from $H_{ext}$. Additionally, all measured $\Delta\sigma_{xy}^{PHE}(H)$ values below 20 K are well fitted by Eq. (2) (Supplementary Fig. 12). The contribution of the PHE coefficients of $a_2$, $a_3$, and $a_4$ to $\Delta\sigma_{xy}^{PHE}(H)$ below 30 K is shown in Fig. 4d. Below 30 K, the dipole induces an $H^2$ dependence of $\Delta\sigma_{xy}^{PHE}(H)$ due to the longitudinal magnetization, as shown in Fig. 1c. In contrast, below 15 K, the contributions of $a_2$, $a_3$, and $a_4$ coexist. While dipolar order still affects $\Delta\sigma_{xy}^{PHE}(H)$, $a_3$ and $a_4$ may

appear due to the OM induced by both the $A_2$- and $T_1$-CMOs. Through theoretical analysis and the PHE results, we confirm that the anomalous behavior of the PHE indeed originates from $A_2$- and $T_1$-CMOs. Our results summarizing the contributions to the AHE and PHE of the cluster magnetic dipole, $A_2$-CMO, and $T_1$-CMO are shown in Table 1 in the Supplementary Materials.

## Discussion

In summary, the PHE of the AFM NIO-227 film exhibits unique features different from those of ferromagnetic materials. The CMOs without magnetization affect the PHE, resulting in higher harmonics of the PHE oscillation beyond the second harmonic. Notably, $A_2$- and $T_1$-CMOs produce fourth and sixth harmonics of PHE oscillations, respectively. This feature of CMOs can be well explained by considering the magnetic characteristic of CMOs called OM in our theoretical calculations. Moreover, the OM of CMOs shows an intriguing $H_{ext}$ dependency, which induces the nonlinear PHE when the angle between $H_{ext}$ and $I$ is 45°.

We would like to make some remarks about the CMOs in AFM materials. According to recent theoretical studies and experiments on antiferromagnets, the magnetic octupole is crucial for stabilizing Weyl fermions near the Fermi energy[18,27,47–51] and inventing magnetic field-free switching spintronic devices[23–25,48–50]. Moreover, AHE, SHE, and Nernst effects have been experimentally observed in kagome antiferromagnets[19,20,25,26,29], in which magnetic structure can also be classified with CMO[18]. Therefore, understanding and manipulating the CMOs are thus important to achieve novel topological phases and physics in AFM systems. However, due to the absence of magnetization, detection and characterization of CMOs and multipoles beyond dipoles have been difficult. In this context, our strategy to distinguish $A_2$- and $T_1$-CMOs could be extensively used to detect and identify cluster multipoles[18,45] via the PHE. Thus, our work paves the way for detecting and identifying AFM order, which is expected to facilitate the development of novel functionalities using AFM materials.

## Methods

### Film fabrication and structural characterization

Fully strained NIO-227 films were grown in situ on insulating YSZ substrates using the "repeated rapid high-temperature synthesis epitaxy" (RRHSE) method[21]. This film growth method is a modified form of pulsed laser deposition, in which repetitive short-term thermal annealing processes are conducted using an infrared laser. The RRHSE consists of two key steps completed in one thermal cycle. In the first step, amorphous stoichiometric NIO-227 and compensational $IrO_2$ layers are deposited with a KrF excimer laser (λ = 248 nm, 5 Hz) at $T$ ~ 600 °C and $P_{O_2}$ ~ 50 mTorr. During the second step, the pyrochlore phase is formed by rapidly raising $T$ to 850 °C at a rate of ~400 °C min$^{-1}$, and the sample is exposed to the high $T$ for a sufficient period. We repeated the deposition and thermal synthesis processes until the desired film thickness was obtained.

### Transport and magnetic properties

Magnetotransport properties were measured via a standard four-point probe method using a commercial physical property measurement system (PPMS; Quantum Design Inc., San Diego, CA, USA), which has a base $T$ of 2 K and a maximum magnetic field of 9 T. Au layer (thickness: ~50 nm) was deposited by an e-beam evaporator on the NIO-227 film and used as an electrode. For the AHE measurement, the current was applied along the $[1\bar{1}0]$ direction, while $H$ was applied in the [111] direction. For the PHE measurement, the current was applied along the $[1\bar{1}0]$ direction and $H$ was rotated between the $[1\bar{1}0]$ and $[11\bar{2}]$ directions. The $R_{xy}$ and longitudinal magnetoresistance $R_{xx}$ below 30 K were simultaneously measured while varying $\phi$ and fixing $H_{ext}$ at ± T. The out-of-plane contribution of $H_{ext}$ caused by a slight misalignment of

the electrode can be removed by symmetrizing the $R_{xy}$ acquired at +9 and −9 T. Then, the planar Hall conductivity $\sigma_{xy}^{PHE}$ is calculated by $\sigma_{xy}^{PHE} = \frac{-\rho_{xy}^{PHE}}{\rho_{xy}^{PHE^2} + \rho_{xx}^2}$ and normalized ($\Delta\sigma_{xy}^{PHE}(\phi)$); here, $-\rho_{xy}^{PHE}$ and $\rho_{xx}$ are the PHE resistivity and longitudinal resistivity, respectively. The detailed symmetrization process of the PHE is provided in Supplementary Materials (Supplementary Fig. 3).

## Data availability

All relevant data presented in this manuscript are available from the authros upon reasonable request. The source data underlying Figs. 2, 3b, c, d, 4b, and Supplementary Fig. 8 are provided as a Source Data file. Source data are provided with this paper.

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

## Acknowledgements

This work was supported by the Research Center Program of the IBS (Institute for Basic Science) in Korea (grant no. IBS-R009-D1). Y.Z. and Y.L. acknowledge the support provided by the Qilu Young Scholars Program of Shandong University. W.J.K. was supported by the U.S. Department of Energy (DOE), Office of Basic Energy Sciences, Division of Materials Sciences and Engineering (contract No. DE-AC02-76SF00515), and the Gordon and Betty Moore Foundation's Emergent Phenomena in Quantum Systems Initiative (grant No. GBMF9072, synthesis equipment). B.-J.Y. was supported by the Samsung Science and Technology Foundation under Project No. SSTF-BA2002-06 and National Research Foundation of Korea (NRF) Grants funded by the Korean government (MSIT) (No. 2021R1A2C4002773 and No. NRF-2021R1A5A1032996).

## Author contributions

J.S. and Y.L. conceived the idea and designed the experiments. T.O. performed the calculation of the orthogonal magnetization effect on the planar Hall effect under the supervision of B.J.Y. J.S. grew and characterized the structure of the samples. J.S., Y.L., Y.Z., E.K., and J.L. performed the magnetotransport measurements. J.S., T.O., W.J.K., B.J.Y., T.W.N., and Y.L. analyzed the results and wrote the manuscript. All authors participated in the discussion during the manuscript preparation.

## Competing interests

## Additional information

**Correspondence and requests** for materials should be addressed to Yangyang Li or Tae Won Noh.

