## [Peer Review File · Nature Communications]

Reviewers' Comments:

Reviewer #1:

Remarks to the Author:

The authors have revealed the existence of different spin texture representations in strained pyrochlore via electrical transport measurements, which provides a new and handy strategy for analyzing complicated magnetic materials. The result is quite interesting, but I still hold a few questions before publication.

1. To my understanding, the AIAO spin texture is formed when compressive strain is applied, and the free energy is composed of contributions from magnetic multipoles. I am curious about the magnetic anisotropy which is seemed to have been obviated in this model. Would the anisotropy be still negligible under strained conditions? Would it affect the analysis of the multipoles? Perhaps a scan of out of plane angle would help to illustrate more about this point.
2. It would become clearer if the authors could quantitatively summarize the proportions of the three spin texture representations: Dipole, A₂ and T₁. It seems that the degrees of freedom provided by curve fitting is more than 3, so it would be possible to figure out the contributions from spin textures.
3. Apart from PHE, is there also angle dependence for the anisotropic magneto resistance?
4. Some figures of the manuscript are unclear, such as figure 1 and figure 2c. For example, the magnetic moments in figure 1 are not clear enough. Therefore, high-resolution images should be given.
5. According to the AHE figure in figure S1, it seems that apart from AHE, topological Hall effect also shows up, especially for the temperature of 4 K and 6 K. Obviously, the curve is not mere AHE. Have the authors analysed this phenomenon? What is the reason for the unusual AHE curve in the film? Does it has relationship with the cluster magnetic multipoles?
6. Minor point. In page 13, 6 line from the bottom, [1-10] direction should not be expressed like this. In page 14, line 10, the citation of Ref. 35 is not normal.

Reviewer #2:

Remarks to the Author:

The authors have measured the planar Hall effect in AFM Nd₂Ir₂O₇, a pyrochlore. At low temperatures, below the Neel temperature T_N, the authors note a periodic behavior of the magnetoresistance, with high Fourier components. They attribute the higher components to magnetic octupolar clusters, and also fit the magnetoresistance vs. magnetic field strength at fixed angle and attribute terms of order H², H³, and H⁴ to dipolar and octupolar order, respectively.

In general, I think the work is carefully done and in principle worthy of publication in Nature Comm. However, the manuscript is not publishable in its present form, and the authors need to respond to my comments and revise the manuscript accordingly.

1. The theoretical analysis (Sec. III of the SI and also "Self-consistent mean-field Hubbard model calculations" in the Methods section leaves a lot out. The Hubbard Hamiltonian Eq. (7) (Eq. (3) in the SI) has a bunch of parameters that are not defined (the vectors d_{ij} , D_{ij} , and R_{ij}) nor are values for them given. Also, the self-consistent calculations indicate that A₂ order exists even in the absence of strain. Does that not mean that a $\sin(4\phi)$ term should show up in the magnetoresistance even in the absence of strain? Also, how are the order parameters A₂ and T₁ defined in terms of the operators of the Hubbard Hamiltonian? I imagine that the results presented in Fig.s4 are expectation values of those operators. Is that correct?
2. Is the temperature dependence the same for the dipolar and octupolar orders? That is, does octupolar order set in at the Neel temperature or is there another temperature scale for them? The authors indicate that the contributions to the Hall conductivity are visible below 15 K. Is that some particular scale or just what the authors observe for no particular reason? Also, why are the octupolar contributions to the Hall resistivity so small? Is there a way to understand that from the Hubbard Hamiltonian?
3. Stylistically, there is a lot of redundancy between the SI and the manuscript (in particular the theoretical discussions in the Methods section and the SI). The authors may want to think about revising them both so that the manuscript reads as a self-contained manuscript and the SI fill in

details.

4. The English, although pretty good, has some missing articles and there are some awkward passages, such as the top of p. 4 "...cluster magnetic octupole (CMO) without magnetization that has the same symmetry as belongs to...."; I think one should say "dipolar order" instead of "dipole order" (top of p. 8). There are other examples - I urge the authors to carefully proofread a revised manuscript and get assistance from a native-English speaker.

5. The authors mention relations between breaking time-reversal symmetry and the appearance of non-trivial topology. In my understanding, a system that respects the combined symmetry operations of time reversal symmetry and any unitary operator cannot have a non-trivial topology. My point is that the text in the manuscript can be construed to say that breaking only TRS is enough to give rise to non-trivial topology, which it is not; the authors should be a little bit more careful in statements relating symmetries and non-trivial topology.

6. I think there is some interesting work by Broholm and Tchernyshyov and co-workers on Mn₃Ge (Physical Review B, 102, 054403, 2020 and references therein) that perhaps indicate similar physics, such as the emergence of a magnetization component perpendicular to an applied field. I think it would be interesting if the authors comment on this class of kagome antiferromagnets (beyond the discussions in Refs. 26 and 28).

Reviewer #3:

Remarks to the Author:

Song et al. describes experimental study on planar Hall effect (PHE) found in antiferromagnetic conductor Nd₂Ir₂O₇, well known for its magnetic structure that can host topological phases. The key finding is that the PHE shows higher harmonics (4th, 6th) in addition to standard second order term as a function of magnetic-field angle. The authors attribute the origin of higher harmonics in PHE to the cluster magnetic octupole that arises in the strained Nd₂Ir₂O₇ films.

The higher order effect in PHE has been theoretically discussed in recent literature, but experimental data has been rather scarce so far. The present data will add new experimental evidence on the higher harmonics PHE. Also, I believe that the use of PHE to potentially probe octupole order in magnetic materials is novel and interesting, and likely would trigger related efforts to unveil magnetic orders by PHE difficult to probe by other means. All this being said, I believe that the manuscript requires some clarifications and serious revisions which are detailed below:

1. The octupole order

What is the independent experimental evidence of octupole order besides PHE?

One peripheral signature for the cluster magnetic octupole seems to be the lack of net magnetization. However, there are no direct magnetization measurements performed on these films. Authors should include magnetic measurements performed, e.g., by SQUID. Since anomalous hall effect clearly shows hysteric behavior normally indicative of net magnetization, it seems important to have direct data on the lack of long-range magnetic order. Even so, I am left wondered what the real experimental evidence of "cluster magnetic octupole (CM)" is in the films reported here at this point.

2. Intro

Introduction contains many general aspects on magnetism and spin ordering, but lacks introduction on how the PHE could help identifying those phases, which readers would be more interested. Authors should consider bringing PHE into a spotlight much earlier in the text to improve readability. In the current version, there are too many detailed discussion on general magnetic order, symmetry breaking, induced topology etc., which are not urgently relevant to understand the authors' work. Besides, first two paragraphs of "Results" section are in fact not results, but again the introduction to magnetic order, which need to be included in the Introduction section. I suggest authors to sharply edit their introduction to highlight core insights that lead to their work, and especially to include the mechanisms how and why PHE can probe the octupole order at least appearing in the second or third paragraph of the introduction.

3.English editing

English editing seems to be necessary. I identified at least a few locations where sentence does not make sense or hard to read, including but not limited to followings:

Page 4: "Particularly, cluster magnetic octupole (CMO) without magnetization that has the same symmetry as belongs to the same irreducible representation as the magnetic dipole."

Page 6: "the OM occurs from the octupolar order couples to the third order of magnetic field in the free energy, which can be used to detect the octupolar order in the system³⁵."

COMMENT TO AUTHOR: Reviewer #1

The authors have revealed the existence of different spin texture representations in strained pyrochlore via electrical transport measurements, which provides a new and handy strategy for analyzing complicated magnetic materials. The result is quite interesting, but I still hold a few questions before publication.

Author's Response:

We greatly acknowledge the reviewer for the positive opinion. We have answered all questions from the reviewer below.

1. To my understanding, the AIAO spin texture is formed when compressive strain is applied, and the free energy is composed of contributions from magnetic multipoles. I am curious about the magnetic anisotropy which is seemed to have been obviated in this model. Would the anisotropy be still negligible under strained conditions? Would it affect the analysis of the multipoles? Perhaps a scan of an out-of-plane angle would help to illustrate more about this point.

Response: We appreciate the constructive question raised by the reviewer.

Before we go into a detailed discussion about the magnetic anisotropy in pyrochlore iridates, we would like to clarify the emergence of AIAO spin texture in pyrochlore iridates. Generally, the AIAO spin texture intrinsically exists in the family of bulk pyrochlore iridates except for $\text{Pr}_2\text{Ir}_2\text{O}_7$ [Matsuhira, K. et al. *J. Phys. Soc. Jpn.* **80**, 094701 (2011)]. For bulk $\text{Nd}_2\text{Ir}_2\text{O}_7$, the AIAO order develops at the Neel temperatures ($T_N^{Ir} \sim 30$ K) [Kim, W. J. et al. *Phys. Rev. B* **98**, 125103 (2018)]. For the thin film, the role of strain in $\text{Nd}_2\text{Ir}_2\text{O}_7$ is to break the crystalline symmetries and tilt the spin configurations [Kim, W. J. et al., *Sci. Adv.* **6.29**, eabb1539 (2020)]. This tilted spin configuration can be considered as the superposition of three cluster multipoles: dipoles, A_2 -octupole, and T_1 -octupole.

The magnetic anisotropy in this system comes from Dzyaloshinskii-Moriya (DM) interaction and single-ion anisotropy. The Ir spins have DM interaction which gives rise to magnetic anisotropy and AIAO spin texture as mentioned above. The Nd spins has strong single-ion anisotropy, because of high spin-9/2. Note that the energy of single-ion anisotropy is proportional to the spin operator square S_z^2 , and single-ion anisotropy for spin-1/2 is absent ($S_z^2 = 1$). The single-ion anisotropy of Nd spins affects Ir spin configuration and the transport by Ir electrons through a strong f - d exchange coupling. We have considered the DM interaction in the Hubbard model in Supplementary Materials. The hopping includes the spin-orbit coupling, which acts as the DM interaction in the large U limit.

We find that the magnetic anisotropy gets stronger below the Nd Neel temperature ($T \sim 15$ K) regardless of strain. The experimental report of $\text{Nd}_2\text{Ir}_2\text{O}_7$ single crystal demonstrates the dramatic change in magnetoresistivity with respect to magnetic field rotation below 15 K [Ueda, K. et al., *Phys. Rev. Lett.* **115**, 056402 (2015)]. Also, the strained $\text{Nd}_2\text{Ir}_2\text{O}_7$ film shows magnetic anisotropy under 15 K. In **Fig. R1**, the out-of-plane magnetic field rotation magnetoconductance exhibits the development of fluctuation below 15 K.

Moreover, we think that the effect of strain on magnetic anisotropy is little through the energy scale analysis. The energy scales that determine the physics in pyrochlore iridate are the energy gap of t_{2g} and e_g orbital (~ 2 eV) and the spin-orbit coupling (~ 0.4 eV), which are very large compared to the bandwidth [Witczak-Krempa, W. et al. *Annu. Rev. Condens. Matter Phys.* **5**, 57–82 (2014)]. According to the previous calculation of bandwidth of strained $\text{Nd}_2\text{Ir}_2\text{O}_7$ film [Kim, W. J. et al., *Sci. Adv.* **6.29**, eabb1539 (2020)], 1 % of compressive strain changes the band energy by less than a few percent of its bandwidth. Thus, magnetic anisotropy is intrinsic in the system.

In addition, we believe that magnetic anisotropy in our free energy model is also considered. The first reason is that the free energy theory considers every possible term that respects the symmetry of the system. Therefore, the free energy theory includes spin-orbit coupling, DM interaction, and single-ion anisotropy since the terms respect the symmetry of the system. Furthermore, the effect of magnetic anisotropy is also considered in the cluster magnetic multipoles. The cluster magnetic multipoles are made up of spin clusters in the unit cell [Suzuki, M. -T. et al. *Phys. Rev. B* **95**, 094406 (2017)]. The spin configurations of magnetic

materials can be classified into cluster magnetic multipoles based on their magnetic point group. In $\text{Nd}_2\text{Ir}_2\text{O}_7$, there are four atoms in the unit cell of pyrochlore iridates, which result in a total of 12 ($= 4 \times 3$) distinct cluster magnetic multipoles [Oh, T. et al. *Phys. Rev. B* **98**, 144409 (2018)]. Since Nd ordering affect the Ir spins by $f-d$ exchange interaction below Neel temperature of Nd (15 K), the affected Ir spin configuration can be expressed in terms of the linear combination of cluster magnetic multipoles. As free energy can contain the coupling of every cluster magnetic multipole to the magnetic field, our free energy model includes the effect of magnetic anisotropy as well. For example, our free energy model provides the fitting function to explain the anisotropy in the out-of-plane rotation magnetoconductance in **Fig. R1**, $\sigma = A_1 \cos 2\theta + A_2 \cos 4\theta + A_3 \cos 6\theta$. The theoretical calculation of out-of-plane rotation magnetoconductivity is following:

We use Onsager's relation for longitudinal conductivity from cluster multipoles.

$$\sigma_{xx} = \sigma_0(H_x^2) + \sigma_1(M_x H_x) + \sigma_2(M_x^2).$$

Let us consider the magnetic field in the $[11\bar{2}]$ plane $\mathbf{H} = H(\cos \phi, 0, \sin \phi)$, the first term is

$$\sigma_{xx}^1 \propto \cos 2\phi,$$

The magnetization from A_2 -octupole (\overrightarrow{M}_{A_2}) is

$$\begin{aligned} M_{A_2,x} &= -\frac{A_2}{2\sqrt{3}} H^2 \sin 2\theta, \\ M_{A_2,y} &= \frac{A_2}{\sqrt{6}} H^2 \cos^2 \theta, \\ M_{A_2,z} &= -\frac{A_2}{4\sqrt{3}} H^2 (-1 + 3 \cos 2\theta), \end{aligned}$$

The magnetization from T_1 -octupole (\overrightarrow{M}_{T_1}) from T_1 -octupole is

$$\begin{aligned} M_{T_1,x} &= \frac{3}{8} H^2 (T_{1x}(-1 + 7 \cos 2\theta) - 2(5\sqrt{2}T_{1y} + 4T_{1z}) \sin 2\theta), \\ M_{T_1,y} &= \frac{3}{8} H^2 ((-3 + 5 \cos 2\theta) T_{1y} - 5\sqrt{2} (1 + \cos 2\theta) T_{1z} - 10\sqrt{2} T_{1x} \sin 2\theta) \end{aligned}$$

$$M_{T_{1,z}} = -\frac{3}{4}H^2(-8 T_{1z} \sin^2 \theta + 4T_{1x} \sin 2\theta + (5\sqrt{2} T_{1y} + 4 T_{1z}) \cos^2 \theta)$$

$\overrightarrow{M_{A_2}}$ from A₂-octupole and $\overrightarrow{M_{T_1}}$ from T₁-octupole give rise to

$$\begin{aligned} \sigma_{xx} = & H^3(x_1 \sin \theta + x_2 \sin 3\theta + x_3 \sin 4\theta + x_4 \sin 5\theta \\ & + x_5 \cos 2\theta + x_6 \cos 4\theta) \\ & + H^4(y_1 \sin \theta + y_2 \sin 3\theta + y_3 \sin 5\theta + y_4 \cos 2\theta + y_5 \cos 4\theta \\ & + y_6 \cos 6\theta). \end{aligned}$$

We added some comments regarding the magnetic anisotropy in cluster multipoles of Nd₂Ir₂O₇ film in our revised manuscript and Supplementary Materials. Please see page 9, lines 9-15 in the revised manuscript, and pages 22-24 in the revised Supplementary Materials.

Figure R1 Temperature dependence of out-of-plane rotation magnetoconductivities. a. Schematic of out-of-plane rotation measurement. The current is applied along $[1\bar{1}0]$ direction and magnetic field is rotated along $[111]$ and $[1\bar{1}0]$ plane. **b.** Experimental magnetoconductivities below 15 K and their fitting. Added as new Supplementary figure s9.

2. It would become clearer if the authors could quantitatively summarize the proportions of the three spin texture representations: Dipole, A₂ and T₁. It seems that the degrees of freedom provided by curve fitting is more than 3, so it would be possible to figure out the contributions from spin textures.

Response: We appreciate the reviewer very much for this thoughtful suggestion. We believe that the way to quantitatively summarize the proportions of the three cluster multipoles is by applying different strengths of strain. The role of strain is to modulate spin configuration in Nd₂Ir₂O₇ film. The modulated spin configuration can be represented by the linear combination of cluster multipoles. In this sense, applying the different strengths of strain can show the modulation level of spin configuration. According to our theoretical calculation, the 1 % compressive strain on Nd₂Ir₂O₇ film can lead to 0.2 % of the dipole, 98.7 % of the A₂ octupole, and 1.10 % of the T₁ octupole (**Fig. R2**). However, the proportion of cluster multipoles in Nd₂Ir₂O₇ film cannot be obtained by the experimental PHE data. That is because the coefficient of each harmonic in PHE can be originated from not only cluster multipoles but also the band structure near the Fermi level [Li, Y. et al. *Adv. Mat.* **33**, 2008528, (2021)]. In this sense, we cannot figure out the exact proportion of cluster multipoles by experimental PHE due to the complexity of the PHE mechanism. We added some comments regarding the proportion of cluster multipoles in Nd₂Ir₂O₇ film in the revised manuscript and Supplementary Materials. Please see page 10, lines 11-13 in the revised main manuscript, and page 32 in Supplementary Materials.

Figure R2 The proportions of cluster multipoles in strained $\text{Nd}_2\text{Ir}_2\text{O}_7$ film. a. The change in proportions of cluster multipoles by various strengths of strain δ . **b.** When strain is 1 % ($\delta = 0.01$), the proportion of cluster dipole (D), A₂ octupole, and T₁ octupole is shown. **Added as new Supplementary figure s11.**

3. Apart from PHE, is there also angle dependence for the anisotropic magneto resistance?

Response: We thank the reviewer for this important question. Following the suggestion, we performed the angle-dependent magnetoresistance measurement. **Fig. R3** is the anisotropic magnetoconductance (AMC) below 30 K. We observed the development of complex features in AMC below 15 K. To understand this feature of AMC, we theoretically calculated the AMC induced by a dipole, A_2 octupole, and T_1 octupole.

We use Onsager's relation for longitudinal conductivity from cluster multipoles.

$$\sigma_{xx} = \sigma_0(H_x^2) + \sigma_1(M_x H_x) + \sigma_2(M_x^2).$$

From $H = H(\cos \phi, \sin \phi, 0)$, the first term is

$$\sigma_{xx}^1 \propto \cos 2\phi,$$

The magnetization from A_2 -octupole ($\overrightarrow{M_{A_2}}$) is

$$M_{A_2,x} = -\frac{A_2}{\sqrt{6}} H^2 \sin 2\phi,$$

$$M_{A_2,y} = -\frac{A_2}{\sqrt{6}} H^2 \cos 2\phi,$$

$$M_{A_2,z} = -\frac{A_2}{2\sqrt{3}} H^2,$$

The $\overrightarrow{M_{A_2}}$ gives rise to

$$\begin{aligned} \sigma_{xx}^{A_2} &= H^3(e_1 \sin \phi + e_2 \sin 3\phi) \\ &+ H^4(f_1 \cos 2\phi + f_2 \cos 4\phi + f_3 \cos 6\phi + f_4 \cos 10\phi), \end{aligned}$$

The magnetization from T_1 -octupole ($\overrightarrow{M_{T_1}}$) from T_1 -octupole is

$$M_{T_{1,x}} = -\frac{3}{4}H^2(2 T_{1x} + T_{1x} \cos 2\phi + (-5\sqrt{2}T_{1z} + T_{1y}) \sin 2\phi),$$

$$M_{T_{1,y}} = \frac{3}{4}H^2(-2 T_{1y} + (5\sqrt{2} T_{1z} + T_{1y}) \cos 2\phi - T_{1x} \sin 2\phi),$$

$$M_{T_{1,z}} = \frac{3}{4}H^2(2 T_{1z} + 5\sqrt{2} T_{1y} \cos 2\phi + 5\sqrt{2} T_{1x} \sin 2\phi)$$

The \vec{M}_T gives rise to

$$\begin{aligned} \sigma_{xx}^{A_2} = & H^3(g_1 \sin \phi + g_2 \sin 3\phi + g_3 \cos 2\phi \\ & + g_4 \cos 4\phi + g_5 \cos 6\phi) \\ & + H^4(h_1 \sin \phi + h_2 \sin 3\phi + h_3 \cos 2\phi \\ & + h_4 \cos 4\phi + h_5 \cos 6\phi + h_6 \cos 8\phi \\ & + h_7 \cos 10\phi). \end{aligned}$$

The AMC induced by cluster multipoles should have a cosine function up to the tenth harmonics. In **Fig. R3**, we fitted the experimental data with our theoretical result very well, which indicates that AMC is also dominated by cluster multipoles. The theoretical calculation of AMC is shown below and added to Supplementary Materials. **Please see page 21, lines 3-9 in the revised Supplementary Materials. We also made some clarification in the revised manuscript, which is highlighted on page 9, lines 9-15 in the main manuscript.**

Figure R3 Anisotropic magnetoconductance and fitting with cluster multipoles contributions. $\sigma_{AMR}(\phi)$ measured at **a.** 2 K, **b.** 4 K, **c.** 6 K, **d.** 8 K, **e.** 10 K, **f.** 12 K, **g.** 15 K, **h.** 20, and **i.** 30 K. The black line is experimental data while the red line is the fitting result with theoretical AMR equation. **Added as new Supplementary figure s8.**

4. Some figures of the manuscript are unclear, such as figure 1 and figure 2c. For example, the magnetic moments in figure 1 are not clear enough. Therefore, high-resolution images should be given.

Response: We thank the reviewer for this important comment. We have replaced figure 1 and figure 2 with high-resolution images. Please see pages 21 and 22 in the revised main manuscript.

5. According to the AHE figure in figure S1, it seems that apart from AHE, topological Hall effect also shows up, especially for the temperature of 4 K and 6 K. Obviously, the curve is not mere AHE. Have the authors analyzed this phenomenon? What is the reason for the unusual AHE curve in the film? Does it have relationship with the cluster magnetic multipoles?

Response: We appreciate the reviewer very much for this detailed comment. Actually, the “hump” feature of AHE in Nd₂Ir₂O₇ film can be explained by contributions of Nd ordering on Ir spin through *f-d* exchange interaction. The bulk Nd₂Ir₂O₇ has two Neel temperatures: Neel temperatures of Ir ($T_N^{Ir} \sim 30$ K) and Nd ($T_N^{Nd} \sim 15$ K) [Kim, W. J. et al. *Phys. Rev. B* **98**, 125103 (2018)]. At each Neel temperature, interesting features of AHE start to appear. First, at $T_N^{Ir} \sim 30$ K, the nonhysteretic AHE behavior appears. Such behavior is known as the absence of domain switching of AIAO order from Ir lattice [Kim, W. J. et al. *Sci. Adv.* **6.29**, eabb1539 (2020)]. The additional hysteretic component starts to appear at $T_N^{Nd} \sim 15$ K. Such hysteretic behavior is due to contribution from Nd order on Ir lattice through *f-d* exchange interaction [Kim, W. J. et al. *Sci. Adv.* **6.29**, eabb1539 (2020)]. This additional hysteretic component results in the antihysteretic behavior and hump feature of AHE below 15 K, which is consistent with the theoretical modeling [Kim, W. J. et al. *Sci. Adv.* **6.29**, eabb1539 (2020)].

Therefore, the AHE of Nd₂Ir₂O₇ film can be fitted with two components (**Fig. R4**): Ir and Nd rather than the topological Hall effect. Hence, we believe that the hump feature below 15 K is due to the additional component from Nd ordering. We added some clarification in the revised main manuscript, which is highlighted on page 6, lines 15-16. Also please see pages 3-4 and Fig. s2 in the revised Supplementary Materials.

Figure R4 Fitting results for the AHE in fully-strained $\text{Nd}_2\text{Ir}_2\text{O}_7$ film. Experimental AHE data (grey circle) and fitting result with Ir (green) and Nd (blue) contribution at **a.** 2 K, **b.** 4 K, and **c.** 6 K. **Added as new Supplementary figure s2.**

6. Minor point. In page 13, 6 line from the bottom, [1-10] direction should not be expressed like this. In page 14, line 10, the citation of Ref. 35 is not normal.

Response: We thank the reviewer for pointing this out. The current direction of transport measurement is changed to $[1\bar{1}0]$. Moreover, the theoretical calculation of PHE in the Method section is removed due to the redundancy of information given in the method section and supplementary materials. Therefore, the miscitation issue of Ref. 35 is solved. Please see page 13, line 18 in the revised main manuscript.

COMMENT TO AUTHOR: Reviewer #2

The authors have measured the planar Hall effect in AFM Nd₂Ir₂O₇, a pyrochlore. At low temperatures, below the Neel temperature T_N , the authors note a periodic behavior of the magnetoresistance, with higher Fourier components. They attribute the higher components to magnetic octupolar clusters, and also fit the magnetoresistance vs. magnetic field strength at fixed angle and attribute terms of order H^2 , H^3 , and H^4 to dipolar and octupolar order, respectively.

In general, I think the work is carefully done and in principle worthy of publication in Nature Comm. However, the manuscript is not publishable in its present form, and the authors need to respond to my comments and revise the manuscript accordingly.

Author's Response:

We thank the reviewer very much for the recommendation. We have answered all the questions raised and revised the manuscript carefully.

1. The theoretical analysis (Sec. III of the SI and also "Self-consistent mean-field Hubbard model calculations" in the Methods section leaves a lot out. The Hubbard Hamiltonian Eq. (7) (Eq. (3) in the SI) has a bunch of parameters that are not defined (the vectors d_{ij} , D_{ij} , and R_{ij}) nor are values for them given. Also, the self-consistent calculations indicate that A_2 order exists even in the absence of strain. Does that not mean that a $\sin(4\phi)$ term should show up in the magnetoresistance even in the absence of strain? Also, how are the order parameters A_2 and T_1 defined in terms of the operators of the Hubbard Hamiltonian? I imagine that the results presented in Fig.s4 are expectation values of those operators. Is that correct?

Response: We thank the reviewer very much for the important comments. As the reviewer addressed, we did not define some parameters like $\vec{d}_{ij}, \vec{D}_{ij}, \vec{R}_{ij}$ that are used in the Hubbard Hamiltonian. We added the definition of those parameters in the revised Supplementary Materials.

Next, we agree with the reviewer that a $\sin 4\phi$ term arises if A_2 -octupole exists in any pyrochlore iridates. Since, the magnetic order of bulk pyrochlore iridates has AIAO, which can be classified as A_2 -octupole [Suzuki, M. -T. et al. *Phys. Rev. B* **95**, 094406 (2017)], the $\sin 4\phi$ term should show up in $\text{Nd}_2\text{Ir}_2\text{O}_7$ even without strain. There is an experimental report that the AIAO order gives rise to $\sin 4\phi$ term [Li, Y. et al. *Adv. Mat.* **33**, 2008528, (2021)]

Lastly, the reviewer asked how the order parameters A_2 -octupole and T_1 -octupole are defined, and whether the results in old Fig. s4 (revised as new Supplementary Fig. s5) are the expectation values of the operators. We define the order parameters as

$$A_2 = \frac{1}{4\sqrt{3}} \langle S_{1x} + S_{1y} + S_{1z} + S_{2x} - S_{2y} - S_{2z} - S_{3x} + S_{3y} - S_{3z} - S_{4x} - S_{4y} + S_{4z} \rangle,$$

$$T_{1x} = \frac{1}{4\sqrt{2}} \langle -S_{1y} - S_{1z} + S_{2y} + S_{2z} + S_{3y} - S_{3z} - S_{4y} + S_{4z} \rangle,$$

$$T_{1y} = \frac{1}{4\sqrt{3}} \langle -S_{1x} - S_{1z} + S_{2x} - S_{2z} + S_{3x} + S_{3z} - S_{4x} + S_{4z} \rangle,$$

$$T_{1z} = \frac{1}{4\sqrt{3}} \langle -S_{1x} - S_{1y} + S_{2x} - S_{2y} - S_{3x} + S_{3y} + S_{4x} + S_{4y} \rangle,$$

where S_{ia} is the spin operator, and $\langle \dots \rangle$ is the expectation value. The results in Fig. s5 are expectation values and we change the axis to $\hat{x}' = [1\bar{1}0], \hat{y}' = [11\bar{2}], \hat{z}' = [111]$. Accordingly, T_1 -octupoles transform as

$$T'_{1x} = \frac{T_{1x} - T_{1y}}{\sqrt{2}}, T'_{1y} = \frac{T_{1x} + T_{1y} - 2T_{1z}}{\sqrt{6}}, T'_{1z} = \frac{T_{1x} + T_{1y} + T_{1z}}{\sqrt{3}},$$

while A_2 -octupole does not transform. We add the definition of order parameters in the Supplementary Materials. **Please see the associated clarifications of order parameters on pages 11-12 and 25 in the revised Supplementary Materials.**

2. Is the temperature dependence the same for the dipolar and octupolar orders? That is, does octupolar order set in at the Neel temperature or is there another temperature scale for them? The authors indicate that the contributions to the Hall conductivity are visible below 15 K. Is that some particular scale or just what the authors observe for no particular reason? Also, why are the octupolar contributions to the Hall resistivity so small? Is there a way to understand that from the Hubbard Hamiltonian?

Response: We thank the reviewer for the constructive comment.

We believe that the ordering temperature of the dipole, A_2 , and T_1 -octupoles is the same as the Neel temperature of Ir (~ 30 K) since all magnetic multipoles are from the Ir magnetic ordering. However, the enlargement of T_1 -octupole occurs at Neel temperature of Nd (~ 15 K) through $f-d$ exchange interaction. Such features of cluster multipoles affect the AHE of strained $\text{Nd}_2\text{Ir}_2\text{O}_7$ film [Kim, W. J. et al. *Sci. Adv.* **6.29**, eabb1539 (2020)], in which different AHE behaviors were observed below Neel temperature of Ir and Nd. While all three multipoles can arise in $\text{Nd}_2\text{Ir}_2\text{O}_7$ film, the magnitude of Ir magnetic moment ($\sim 0.2 \mu_B$) and induced T_1 -octupole are small. Meanwhile, when the temperature is decreased below the Neel temperature of Nd (~ 15 K), Nd orders induce an additional effective field on Ir lattice through $f-d$ exchange interaction. This additional contribution amplifies the magnitude of T_1 octupole in $\text{Nd}_2\text{Ir}_2\text{O}_7$ film. The amplified magnitude T_1 octupole induces spontaneous AHE in $\text{Nd}_2\text{Ir}_2\text{O}_7$ film [Kim, W. J. et al. *Sci. Adv.* **6.29**, eabb1539 (2020)]. Indeed, in our work, we also show the coherent behavior with AHE in PHE measurements. The second and fourth harmonics of PHE that were induced by dipole and A_2 octupole start to appear below 30 K. On the other hand, the sixth harmonics of PHE induced by amplified T_1 octupole become more visible below 15 K due to $f-d$ exchange interaction.

We explain the relationship between octupolar orders and the Hall conductivity. First, we here address the relation of octupolar orders to anomalous Hall conductivity. We recall that there are two kinds of octupolar orders in the strained $\text{Nd}_2\text{Ir}_2\text{O}_7$ thin films, A_2 and T_1 -octupoles. There are several differences between A_2 and T_1 -octupoles. First, A_2 -octupole is a scalar order parameter, while T_1 -octupole is a vector one. Second, the magnetic point group for A_2 -octupole is $-4'3m'$, while that for T_1 -octupole is $-42'm'$. Note that the magnetic point group of dipoles (the ferromagnetic orders) is $-42'm'$. Since the magnetic point group of dipoles and

T_1 -octupoles are the same, T_1 -octupole can give rise to anomalous Hall Effect [Suzuki, M. –T. et al. *Phys. Rev. B* **95**, 094406 (2017)]. A_2 -octupole does not give rise to the anomalous Hall Effect since it has a different magnetic point group.

The contribution of A_2 -octupole on anomalous Hall conductivity can be also understood by the Hubbard Hamiltonian of this system. Without magnetic ordering, the Hamiltonian has a quadratic band crossing at the Γ point. When A_2 -octupole sets in, the quadratic band crossing is broken into 4 pairs of Weyl nodes that are exactly at the Fermi level and along the $\Gamma - L$ line. The anomalous Hall conductivity is proportional to the distance between a pair of Weyl nodes [Burkov, AA. et al. *Phys. Rev. Lett.* **113**, 187202 (2014)]. Since the sum of distances between 4 pairs of Weyl nodes is canceled out, anomalous Hall conductivity vanishes. When A_2 -octupole gets increased, 4 pairs of Weyl nodes move toward L points. When they reach the L point, the system becomes insulating since Weyl nodes are pair-annihilated. Since this is a trivial insulator, the anomalous Hall conductivity should vanish [Oh, T. et al. *Phys. Rev. B* **98**, 144409 (2018)].

As for why the octupolar contributions to the planar Hall conductivity are small. We calculated the magnetization induced by strain and magnetic field in the octupolar system $Nd_2Ir_2O_7$ (**Fig. R5**). We recall that the planar Hall conductivity in $Nd_2Ir_2O_7$ is generated by the orthogonal magnetization that results from the coupling between the octupolar orders (higher-rank multipoles) and the magnetic field. Since orthogonal magnetization is the coupling between the higher rank multipoles and magnetic field, it should have small values. As expected (**Fig. R5b**), the magnetization (orthogonal magnetization) induced by the magnetic field in the strained $Nd_2Ir_2O_7$ is very small (at the scale of $10^{-3} \mu_B/\text{atom}$). Hence, the octupolar contributions to the planar Hall conductivity are small. We added some clarification in the revised main manuscript and Supplementary Materials, which is highlighted on page 9, lines 9-15 in the revised main manuscript, and pages 5-6 and 19-20 in Supplementary Materials.

Figure R5 Magnetization induced by strain and magnetic field. **a.** The change of magnetization when the strain δ varies from 0 to 0.01 with $\vec{B} = 0$ **b.** The change of magnetization when magnetic field strength varies from 0 to 0.02 with $\phi = 0, \delta = 0.005$. Added as new Supplementary Figure s5c,d.

3. Stylistically, there is a lot of redundancy between the SI and the manuscript (in particular the theoretical discussions in the Methods section and the SI). The authors may want to think about revising them both so that the manuscript reads as a self-contained manuscript and the SI fill in details.

Response: We thank the reviewer for pointing this out. We have removed all redundant details in the Methods section. Especially, we found that the theoretical calculation of orthogonal magnetization and PHE from cluster magnetic octupoles in Methods and Supplementary materials are similar. Therefore, we decide to remove the theoretical description of them in the Methods section. **Please see the Method section in the revised manuscript.**

4. The English, although pretty good, has some missing articles and there are some awkward passages, such as the top of p. 4 "...cluster magnetic octupole (CMO) without magnetization that has the same symmetry as belongs to..."; I think one should say "dipolar order" instead of "dipole order" (top of p. 8). There are other examples - I urge the authors to carefully proofread a revised manuscript and get assistance from a native-English speaker.

Response: We thank the reviewer for pointing this out. The awkward passage regarding the symmetry of CMO is corrected. Also, we have changed all of the dictation regarding magnetic ordering to “dipolar” and “octupolar” orderings. Furthermore, we also proofread the manuscript by both native-English speaker and Springer Nature English Services. Please see associated changes on page 3, line 19-21, page 7, line 4, page 7, line 11, page 10, line 20, and page 24, line 8 in the revised main manuscript.

5. The authors mention relations between breaking time-reversal symmetry and the appearance of non-trivial topology. In my understanding, a system that respects the combined symmetry operations of time reversal symmetry and any unitary operator cannot have a non-trivial topology. My point is that the text in the manuscript can be construed to say that breaking only TRS is enough to give rise to non-trivial topology, which it is not; the authors should be a little bit more careful in statements relating symmetries and non-trivial topology.

Response: We thank the reviewer for this important comment. We agree with the reviewer that our previous statement is not accurate. Therefore, we have revised the statements relating to symmetries of CMO to “In particular, some cluster magnetic octupole (CMO) (i.e. T_1 octupole) belongs to same magnetic point group ($-42'm'$) as the magnetic dipole, leading to symmetry breaking and generating nonvanishing Berry curvature.” We thank the reviewer again for this important comment. The revised statement can be found on **page 3, lines 19-21 in the revised manuscript.**

6. I think there is some interesting work by Broholm and Tchernyshyov and co-workers on Mn₃Ge (Physical Review B, 102, 054403, 2020 and references therein) that perhaps indicate similar physics, such as the emergence of a magnetization component perpendicular to an applied field. I think it would be interesting if the authors comment on this class of kagome antiferromagnets (beyond the discussions in Refs. 26 and 28). Mn₃Ge

Response: We thank the reviewer for the good references that we missed. We agree with the reviewer that the work by Broholm and Tchernyshyov has similar physics about the perpendicular magnetization in kagome antiferromagnets [Chen. Y. et al. *Physical Review B*, **102**, 054403 (2020)]. According to experimental work on the torque magnetometry of Eu₂Ir₂O₇ [Liang, T. et al. *Nature Physics* **13**, 599-603 (2017)], orthogonal magnetization should exist under the presence of cluster magnetic octupoles. Also, the magnetic materials can be classified with cluster magnetic multipoles based on their magnetic point group [Suzuki, M. -T. et al. *Phys. Rev. B* **95**, 094406 (2017)]. The spin structure of kagome antiferromagnet Mn₃X (X = Sn, Ge) can also be classified with cluster magnetic octupole. In this sense, cluster magnetic multipoles should affect the transport phenomena of Mn₃X (X = Sn, Ge). For example, AHE, spin Hall effect, and Nernst effect have been experimentally observed in kagome antiferromagnet Mn₃X (X = Sn, Ge) [Kiyohara. N. et al. *Phys. Rev. Applied* **5**, 0604009 (2016), Iklas. M. et al. *Nat. Phys.* **13**, 1085 (2017), Zhang. Y. et al. *Phys. Rev. B* **95**, 075128 (2017), and Nayak. A. K. et al. *Sci. Adv.* **e1501870** (2016)]. While these anomalous transport phenomena were elucidated by the nonvanishing Berry curvature mechanism, the cluster magnetic multipoles in kagome antiferromagnets are believed to play a crucial role.

In this regard, we made some comments about this point and added them to the revised manuscript. Also, we added the work of Mn₃Ge by Broholm and Tchernyshyov and related works of kagome antiferromagnets as reference numbers 19, 20, 25, 26, and 29. Please see page 3, lines 23, page 4, line 1, and page 11, lines 17-19 in the revised main manuscript.

COMMENT TO AUTHOR: Reviewer #3

Song et al. describes experimental study on planar Hall effect (PHE) found in antiferromagnetic conductor Nd₂Ir₂O₇, well known for its magnetic structure that can host topological phases. The key finding is that the PHE shows higher harmonics (4th, 6th) in addition to standard second order term as a function of magnetic-field angle. The authors attribute the origin of higher harmonics in PHE to the cluster magnetic octupole that arises in the strained Nd₂Ir₂O₇ films.

The higher order effect in PHE has been theoretically discussed in recent literature, but experimental data has been rather scarce so far. The present data will add new experimental evidence on the higher harmonics PHE. Also, I believe that the use of PHE to potentially probe octupole order in magnetic materials is novel and interesting, and likely would trigger related efforts to unveil magnetic orders by PHE difficult to probe by other means. All this being said, I believe that the manuscript requires some clarifications and serious revisions which are detailed below:

Author's Response:

We thank the reviewer very much for the positive evaluation. We have answered all the questions raised and revised the manuscript carefully.

1. The octupole order

What is the independent experimental evidence of octupole order besides PHE?

One peripheral signature for the cluster magnetic octupole seems to be the lack of net magnetization. However, there are no direct magnetization measurements performed on these films. Authors should include magnetic measurements performed, e.g., by SQUID. Since anomalous hall effect clearly shows hysteretic behavior normally indicative of net

magnetization, it seems important to have direct data on the lack of long-range magnetic order. Even so, I am left wondering what the real experimental evidence of “cluster magnetic octupole (CM)” is in the films reported here at this point.

Response: We appreciate the reviewer very much for this important comment. We agree with the reviewer that peripheral evidence of cluster octupole is the lack of magnetization. Following the suggestion, we performed the magnetization measurement. As expected, the magnetization of strained $\text{Nd}_2\text{Ir}_2\text{O}_7$ film at 2 K exhibits a lack of magnetization. However, the finite AHE at 2K is observed (**Fig. R6**). This feature of AHE in strained $\text{Nd}_2\text{Ir}_2\text{O}_7$ film is due to nonvanishing Berry curvature induced by T_1 -octupole [Kim, W. J. et al. *Sci. Adv.* **6.29**, eabb1539 (2020)].

We believe that the appearance of the fourth and sixth harmonics of PHE in $\text{Nd}_2\text{Ir}_2\text{O}_7$ film in specific temperatures is the experimental evidence of cluster magnetic octupoles. Both A_2 and T_1 octupole are antiferromagnetic ordering, in which magnetization is absent. This leads to difficulty in probing them. Indeed, the method to detect the AIAO (A_2 octupole) spin structure of $\text{Nd}_2\text{Ir}_2\text{O}_7$ has been limited to the neutron scattering technique [Guo, H. et al. *Physics. Rev. B* **94**, 161102(R) (2016)]. Moreover, the magnetic spin ordering in thin films cannot be detected by the neutron scattering technique since the film is too thin (the mass of the film is not enough for the neutron scattering). In contrast, the planar Hall effect can be performed in thin film samples. Importantly, under a magnetic field, each cluster multipole exhibits distinctive harmonics of PHE oscillation in our $\text{Nd}_2\text{Ir}_2\text{O}_7$ film. Therefore, we believe that the PHE can be the practical way to distinguish and identify antiferromagnetic cluster octupoles.

To clarify the understanding of our findings, we added comments and magnetization data in the revised main manuscript and Supplementary Materials. Please see page 6, lines 15-16 in the revised main manuscript and Fig. s1c in the Supplementary Materials.

Figure R6. Magnetization and AHE at 2K in $\text{Nd}_2\text{Ir}_2\text{O}_7$ film. Added as Supplementary Figure s1c.

2.Intro

Introduction contains many general aspects on magnetism and spin ordering, but lacks introduction on how the PHE could help identifying those phases, which readers would be more interested. Authors should consider bringing PHE into a spotlight much earlier in the text to improve readability. In the current version, there are too many detailed discussions on general magnetic order, symmetry breaking, induced topology etc., which are not urgently relevant to understand the authors' work. Besides, first two paragraphs of "Results" section are in fact not results, but again the introduction to magnetic order, which need to be included in the Introduction section. I suggest authors to sharply edit their introduction to highlight core insights that lead to their work, and especially to include the mechanisms how and why PHE can probe the octupole order at least appearing in the second or third paragraph of the introduction.

Response: We thank the reviewer for this constructive comment. We have revised the introduction of the manuscript with the constructive suggestion given by the reviewer. The discussions about antiferromagnetic order, spintronics, and topology are shortened and the new paragraph about the planar Hall effect to give spotlight is added to the third paragraph of the introduction. The added paragraph is following:

"The planar Hall effect (PHE) has been considered as a method of probing the physical properties of materials such as magnetism and topology. The PHE corresponds to the development of a Hall voltage when electric and magnetic fields are coplanar, which is different from the usual Hall effect where they are perpendicular to each other. Initially, the PHE was observed in ferromagnetic systems, detecting the anisotropic magnetization of ferromagnetic materials. Additionally, the PHE has recently been in the spotlight due to its role in detecting topological characteristics such as the chiral anomaly arising from Weyl points in Weyl semimetals³⁶⁻³⁹. The associated PHE in both ferromagnets and Weyl semimetals exhibit $\sin(2\phi)$ or second harmonic PHE oscillations. Recently, higher harmonics PHE oscillations beyond second have been reported in topological systems⁴¹⁻⁴⁴, which indicates that the PHE can be a practical tool to probe additional hidden parameters in materials such as CMOs in a system."

Lastly, “Result” is reorganized by starting the section with a discussion of sample characterization. Please see the associated changes on pages 3, 4, and 5 in the revised main manuscript. We would like to thank the reviewer again for this constructive comment and believe that the quality of the manuscript is greatly enhanced.

3.English editing

English editing seems to be necessary. I identified at least a few locations where sentence does not make sense or hard to read, including but not limited to followings:

Page 4: “Particularly, cluster magnetic octupole (CMO) without magnetization that has the same symmetry as belongs to the same irreducible representation as the magnetic dipole.”

Page 6: “the OM occurs from the octupolar order couples to the third order of magnetic field in the free energy, which can be used to detect the octupolar order in the system³⁵.”

Response: We thank the reviewer for pointing this out. The confusing passages regarding the symmetry of CMO and OM have been corrected. The revised sentences are following:

Page 4: “Particularly, cluster magnetic octupole (CMO) without magnetization that has the same symmetry as belongs to the same irreducible representation as the magnetic dipole.”

→“ In particular, some cluster magnetic octupole (CMO) (i.e. T_1 octupole) belongs to the same magnetic point group ($-42'm'$) as the magnetic dipole, leading to symmetry breaking and generating nonvanishing Berry curvature.”

Page 6: “the OM occurs from the octupolar order couples to the third order of magnetic field in the free energy, which can be used to detect the octupolar order in the system³⁵.”

→ Based on comment # 2 from Reviewer 3, we have revised the introduction of the manuscript, and the detailed description of orthogonal magnetization has been removed.

Furthermore, we have proofread not only the points mentioned by the reviewer but also the entire manuscript carefully by Springer Nature English Services. **Please see the associated changes on page 3, lines 19-22 in the revised manuscript.**

Reviewers' Comments:

Reviewer #1:

Remarks to the Author:

The authors address my previous comments and questions very well. I also found the other two referees evaluate the work positively. Thus I recommend its publication as it is.

Reviewer #2:

Remarks to the Author:

The authors have responded satisfactorily to my comments, and I can now recommend the manuscript for publication

Reviewer #3:

Remarks to the Author:

Authors have successfully responded to my earlier comments and revised the manuscript accordingly. I believe the use of higher order PHE in detecting magnetic octupole order represents a novel path to condensed matter research and the quality of the experimental data presented in the manuscript is high.

REVIEWERS' COMMENTS

Reviewer #1 (Remarks to the Author):

The authors address my previous comments and questions very well. I also found the other two referees evaluate the work positively. Thus I recommend its publication as it is.

Response: We thank the reviewer for the recommendation.

Reviewer #2 (Remarks to the Author):

The authors have responded satisfactorily to my comments, and I can now recommend the manuscript for publication

Response: We thank the reviewer for the recommendation

Reviewer #3 (Remarks to the Author):

Authors have successfully responded to my earlier comments and revised the manuscript accordingly. I believe the use of higher order PHE in detecting magnetic octupole order represents a novel path to condensed matter research and the quality of the experimental data presented in the manuscript is high.

Response: We thank the reviewer for the recommendation